# Biosensing Chlorpyrifos in Environmental Water Samples by a Newly Developed Carbon Nanoparticle-Based Indirect Lateral Flow Assay

**DOI:** 10.3390/bios12090735

**Published:** 2022-09-07

**Authors:** Linda Willemsen, Jan Wichers, Mang Xu, Richard Van Hoof, Coby Van Dooremalen, Aart Van Amerongen, Jeroen Peters

**Affiliations:** 1Wageningen Food Safety Research, 6708 WB Wageningen, The Netherlands; 2Wageningen Food & Biobased Research, 6708 WG Wageningen, The Netherlands; 3Wageningen Plant Research, 6708 PB Wageningen, The Netherlands

**Keywords:** chlorpyrifos, lateral flow, immunoassay, surface water, on-site, environment, honey, bees

## Abstract

Pesticides are used in agriculture to prevent pests. Chlorpyrifos (CHLP) is an insecticide with potentially detrimental effects on humans, bees, and the aquatic environment. Its effects have led to a total ban by the European Union (EU), but outside the EU, CHLP is still produced and used. An indirect lateral flow immunoassay (LFIA) for the detection of CHLP was developed and integrated into a cassette to create a lateral flow device (LFD). Species-specific reporter antibodies were coupled to carbon nanoparticles to create a detector conjugate. Water samples were mixed with a specific CHLP monoclonal antibody and detector conjugate and applied to the LFD. Dose-response curves elicited the detection of low concentrations of CHLP (<1 µg/L). This sensitivity was recorded through a rapid handheld digital imaging device but also visually by naked eye. The CHLP LFD was applied to a range of European surface water samples, fortified with CHLP, revealing a sensitivity in these matrices of 2 µg/L, both by digital and visual analysis. To improve the simplicity of the CHLP LFIA, the assay reagents were dried in tubes, enabling to carry out the test by simply adding water samples and inserting the LFIA strips. This CHLP LFIA is thus suited for the on-site screening of surface waters.

## 1. Introduction

Pesticides are used in agriculture for pest control. These pesticides can be divided into several subclasses, such as insecticides, fungicides, and herbicides. Their usage in agriculture and their accumulation in the environment have always been controversial and their environmental impacts have been extensively assessed [1]. The effect of pesticide-free urban green spaces on the other hand has also been critically studied [2]. Chlorpyrifos (O,O-diethyl O-3,5,6-trichloro-2-pyridyl phosphorothioate) is an organophosphate insecticide that was introduced to the market in 1965 [3] and is also known by its trade names Dursban and Lorsban [4]. It has been effectively applied to target the pest-insects of citrus trees, onions, corn, and seeds amongst others [5,6]. In humans, chlorpyrifos (CHLP) can cause cardiovascular diseases, endocrine disruption, neurotoxic effects and can even be lethal at high doses [7,8,9]. CHLP has a strong impact on the aquatic environment, as it is not only neurotoxic via inhibiting the cholinesterase (ChE) enzyme activity, but also genotoxic to freshwater fish [10]. More recent studies showed endocrine and neurodevelopment disrupting effects of CHLP on *Xenopus laevis* brain formation [11]. Additionally, CHLP is highly toxic to bees, through oral exposure (oral LD_50_ 360 ng/bee), or even more so via direct contact (contact LD_50_ 70 ng/bee) [12]. CHLP has been found in pollen, honey, beeswax, and honeybee individuals [13,14,15,16]. Adult bees exposed to sub-lethal levels of CHLP showed reduced walking, increased grooming, impaired righting reflexes, unusual abdominal spasms [17], morphological deviations to the body and mouthparts [18], and reduced appetitive learning and specificity of memory recall [19]. As CHLP is also found in surface waters and dew, water may impose a threat to bees next to foraging on contaminated flowers [20,21,22].

A study, based on food monitoring and human biomonitoring, showed strong reductions of CHLP usage from 2016 onwards, which can be ascribed to regulations and awareness of the potentially harmful effects of CHLP [23]. In addition, the production of CHLP by Dow Chemical/Coverta was halted [9]. Nevertheless, since CHLP is not patent-protected, it is still produced in several countries, including India and China [9], and although pesticides may be banned by global institutions, it does not mean that they are not in use anymore. In a few developing countries, banned pesticides are still reported to be illegally used [24]. Within Europe, the illegal use of CHLP on flowering trees in Austria led to the complete extinction of more than 50 honeybee colonies [25]. In 2020, the EU decided to impose a total ban on CHLP and set a low maximum residue level (MRL) of 0.01 mg/kg as the default value for all products [26,27]. This means that fast screening methods need to be sensitive, both for products and water [10,28,29,30,31]. For monitoring the presence of CHLP in the environment, analytical instrumental analysis is most applied, e.g., multi-detection methods for pesticides, including CHLP [32,33]. These analytical multi-detection methods are highly reliable, but in general also expensive, time-consuming, not portable and suited for on-site use, and require highly skilled personnel for sample preparation, machine operation, and data analysis.

A range of less expensive and more rapid methods have been developed for the detection of CHLP as well. Zhu et al. combined surface-enhanced Raman spectroscopy and chemometrics for the rapid detection of CHLP in tea samples after QuEChERS-based sample extractions and with ppm (mg/kg) based sensitivities [34]. Even though this method may already be considered as rapid, the sample extraction procedure is still not suitable for rapid point of need approaches. Sankar et al. developed a paper-based device for the rapid detection of CHLP in water samples based on the inhibition of lipase enzymatic activity, enabling on-site use [35]. This assay was combined with smartphone readout and had an incubation time of 15 min, but the limit of quantification of 200 µg/L is rather high. Different formats of enzyme-linked immunosorbent assay (ELISA)-based immunoassays are also popular tools for the detection of a wide range of contaminants, including CHLP [36,37]. Despite being robust and semi-high throughput, their portability and suitability for rapid testing are rather low. Another well-known immunoassay format for rapid and point-of-need testing is the lateral flow immunoassay (LFIA) [38]. Today, this diagnostic platform is also known to the general public, as lateral flow tests have been substantially used worldwide for self-testing during the ongoing SARS-CoV-2 pandemic [39]. Both viral antigens in nasopharyngeal swabs and specific antibodies in the blood can be easily and rapidly detected. Besides large proteins, such as viral antigens and antibodies, the LFIA also qualifies for the detection of small molecules, such as phytoproducts, mycotoxins, and pesticides [40,41,42,43]. An LFIA for the direct detection of CHLP based on colloidal gold detection has been developed previously [44].

In the present study, a novel indirect LFIA was developed for the detection of CHLP, implementing Goat anti-Mouse IgG antibody molecules (GAM) immobilized onto carbon nanoparticles (CNP) as the detector conjugate (GAM-CNP). The readout of the results was carried out by rapid intensity-based imaging utilizing a handheld detector (Cube) and visually by the naked eye. The sensitivity with a pure standard in buffer was below 1 µg/L, both by the Cube and naked eye. This CHLP LFIA was then applied to tap water, aquaponics water, and six independent European surface water samples, revealing a sensitivity of 2 µg/L in these matrices. For future point-of-need applicability, the LFIA was designed for implementation into cassettes with sample and assay reagents presented in microtiter plate wells and eventually a simplified approach of the LFIA, i.e., implementing ready-to-use reagent tubes, showed good perspectives for the easy on-site detection of CHLP in water samples with high sensitivity.

## 2. Materials and Methods

### 2.1. Instruments

For spraying and cutting the membranes, a XYZ 3060 BioDot Dispense Platform and a CM4000 Biodot Guillotine (BioDot Inc., Irvine, CA, USA) were used. The readout of LFIAs was performed on a Cube analyzer running on confirmation software, version V1.5.062 (Chembio, Berlin, Germany). Images of the LFIA results were recorded with a Samsung A50 smartphone and adjusted for white balance with Adobe Photoshop software (23.4.1 release), using the curves option and the wicking pad as the reference. The sonication of the CNP conjugate was performed in a sonicator bath (VWR International, Radnor, Pennsylvania, NC, USA) and the Bioruptor Plus Diagenode ultrasonicator (SA, Seraing, Belgium). Measuring antibody concentrations was performed by a DS-11 FX spectrophotometer (DeNovix, Wilmington, NC, USA). For centrifugation steps, the Sigma 2K15 centrifuge was used (Sigma Laborzetrifugen GmbH, Osterode am Harz, Germany). To dry the assay reagents in glass tubes, we used the reacti-therm^TM^ heating and stirring modules from Thermo Fisher (Rockford, IL, USA) combined with a nitrogen gas flow.

### 2.2. Chemicals and Materials

The CHLP monoclonal antibody (mAb) (AT2791, 5.50 mg/mL) and chlorpyrifos-bovine serum albumin (BSA) conjugate (CHLP-BSA) (AG279, 5.70 mg/mL) were purchased from Ecalbio (Wuhan, China). The cross-reactivity of other pesticides with the CHLP mAb were previously tested in a multiplex microsphere immunoassay format within the H2020 B-GOOD project, focusing on pesticides harmful to bees. In total, 88 pesticides were tested from the pesticide classes neonicotinoids, pyrethroids, avermectins, carbamates, organophosphates, and some single pesticides, including fipronil. From these pesticides, only dicrotophos, triazophos, parathion, and methyl-parathion showed significant cross-reactivity but only at relatively high concentrations (1000 µg/mL) (data not shown). Chlorpyrifos-ovalbumin conjugate (CHLP-OVA), 1 mg/mL, was kindly provided by Dr. Yirong Guo of Zhejiang University. Goat anti-Mouse IgG FcY (GAM) specific polyclonal antibodies (1.8 mg/mL, 115-005-071) and the donkey anti-Goat IgG (DAG) polyclonal (H+L) antibodies (1.3 mg/mL, 705-005-003) were obtained from Jackson ImmunoResearch (Sanbio, Uden, The Netherlands). The CHLP stock solution (2000 µg/mL in MeOH) was kindly supplied by the pesticides department of the Wageningen Food Safety Research (WFSR). LFIA reagents were sprayed on CN140 nitrocellulose (NC) membranes (Unisart, Sartorius, Gottingen, Germany). Plastic backing cards, sample pads, and aluminum foil pouches for storage of the produced LFIAs, were purchased from Kenosha (Amstelveen, the Netherlands). Wicking pads were purchased from Whatman (GE Healthcare, Eindhoven, the Netherlands) and Minipax absorbent packets from Merck (Darmstadt, Germany) Global surface water samples from rivers, ponds, and brooks had been previously collected by Dr. Rubing Zou [45] and were supplemented with additional samples collected by WFSR. Aquaponics water was kindly provided by the Circle (Rome, Italy) (Appendix A). Boric acid, Tween-20, and sodium azide (NaN_3_) were purchased from Merck (Darmstadt, Germany); BSA, phosphate-buffered saline (PBS) tablets, Triton X-100 and tergitol from Sigma-Aldrich (Zwijndrecht, The Netherlands); and sodium tetraborate from VWR (VWR, Leuven, Belgium). Deionized water (>18.2 MΩ/cm) was freshly prepared by filtering distilled water through a Milli-Q direct water purification system (Millipore, Burlington, MA, USA) for the preparation of buffers and solutions. The dilutions of the CHLP stock were prepared in Methanol Ultra LC-MS from Actuall Chemicals (Oss, the Netherlands). Spot-based LFIAs were developed in cellstar 96-well plates (Greiner bio-one, Alphen a/d Rijn, the Netherlands), cassettes for the construction of the lateral flow device (LFD) were kindly supplied by Zhejiang University, and zeba spin columns (7K MWCO), used for desalting antibody solutions, were obtained from Thermo Scientific (Rockford, IL, USA). The conjugation of GAM to amorphous CNPs was performed using spezial Schwartz 4 CNPs (Orion Engineered Carbons GmbH, Eschborn, Germany), which were previously characterized [46]. The AR glass tubes 75 × 11.5 × 0.7 mm were obtained from VOS instrumenten b.v. (Zaltbommel, the Netherlands).

### 2.3. Conjugation of GAM Antibodies to Amorphous CNPs

The conjugation of the GAM polyclonal antibody (pAb) to CNPs was performed as described previously by Sharma et al. [47]. In short, a 1% (*w*/*v*) aqueous suspension of amorphous CNPs was sonicated in an ultrasonic bath for 1 h at 40 kHz at room temperature (RT). Subsequently, a 0.2% (*w*/*v*) CNP suspension was prepared in 5 mM Borate buffer (BB) (pH 8.8) containing boric acid and sodium tetraborate. This suspension was sonicated for 5 min using the ultrasonic bath. For conjugation, the purified GAM pAb was desalted/buffer exchanged against 5 mM BB (pH 8.8) using 0.5 mL spin columns. One mL of the 0.2% (*w*/*v*) carbon suspension was mixed with the purified antibody preparation to a final antibody concentration of 350 µg/mL. The content was stirred at 4 °C for 12 h on a magnetic stirrer. Subsequently, 1/10 volume of washing buffer (5 mM BB, pH 8.8 containing 1% BSA) was added and incubation was prolonged for one hour. The suspension was subsequently centrifuged at 13,636× *g* (4 °C, 15 min). The pellet was washed three times with washing buffer and finally suspended in storage buffer (100 mM BB, pH 8.8 containing 1% BSA) to a final concentration of 0.2% (*w*/*v*) CNPs. The GAM-CNP conjugate was stored at 4 °C until further use. The reproducibility, reliability, and efficiency of this coupling procedure have been proven by quantitative computer image analysis and was reported previously [48].

### 2.4. Method Setup by Implementation of Spot-Based Strips

The initial set-up and optimization of the CHLP LFIAs was performed by spotting the CHLP-BSA, or the CHLP-OVA conjugate, and the DAG pAb manually onto the NC. To this end, the NC membranes (30 cm width, 2.5 cm length) were attached to a plastic backing card and overlaid with a 15 mm wicking pad. This assembled NC card was then cut into 4 mm strips. The stock solution of CHLP-BSA and the CHLP-OVA, both in PBS, were diluted in PBS to obtain a concentration series of 1 mg/mL, 0.75 mg/mL, 0.5 mg/mL, 0.25 mg/mL, and 0.125 mg/mL. The DAG mAb was diluted to 0.25 mg/mL in 5 mM BB (pH 8.8). CHLP-BSA (or CHLP-OVA) and the DAG pAb were spotted manually with a micropipette, dispensing 0.5 µL onto the NC membrane, 5 mm part from each other, with the DAG pAb last in order of the flow direction. The strips were then dried at RT for 30 min and stored at RT in a sealed package containing Minipax absorbent packets, until further use. Next, the dilution series of the CHLP mAb stock (1:100–1:512,000) were prepared in running buffer (RB) composed of 0.01 M PBS pH 7.4, 1% BSA (*w*/*v*), and 0.05% Tween-20 (*v*/*v*). CHLP calibration standards were prepared by 10-fold stepwise dilutions, resulting in a concentration range of 0.1 to 100 µg/L. Diluted CHLP mAb (1 µL) and GAM-CNP conjugate (1 µL) were added to the diluted CHLP antigen (98 µL), mixed and then added to a well of a 96-well plate. The same procedure was repeated for the following running buffers: RB2: 100 mM BB, 1% BSA (*w*/*v*), 0.05% Tween-20 (*v*/*v*), and 0.01% Tergitol (*v*/*v*); RB3: 100 mM BB, 1% BSA (*w*/*v*), 0.2% Tween-20 (*v*/*v*), and 0.05% Triton X-100 (*v*/*v*); RB4: 100 mM BB, 1% BSA (*w*/*v*) and 0.05% Tween-20 (*v*/*v*). The LFIA strips were placed upright in a well and incubated for 10 min. After incubation, the strips were placed on a white sheet of paper and photographed (using daylight for exposure).

### 2.5. Preparation of Line-Based Strips for Sensitivity Testing

For the production of line-based LFIAs, the test line with the CHLP-BSA conjugate was sprayed onto the NC membranes at 0.125 mg/mL and the DAG pAb at 0.05 mg/mL using 0.01M PBS (for CHLP-BSA) and 5 mM BB (for DAG pAb) as the spraying buffers. The sprayed NC was dried for 30 min at RT. Next, the dried NC was secured on a plastic backing and overlaid with a wicking pad (as described previously in paragraph 2.4) at the top end and a sample pad at the bottom end. Subsequently, the assembly was cut into 4 mm strips by the guillotine. The cut strips were placed into a dedicated cassette and stored at RT.

### 2.6. Sensitivity Testing of the CHLP LFIAs

CHLP calibration standards of 100, 50, 10, 5, 1, and 0.5 µg/L in RB were prepared to test the sensitivity of the developed LFD. The negative control was the RB buffer. The CHLP mAb was diluted 1:3200 in RB and 1 µL was added to the diluted CHLP standards and negative control (98 µL). Next, 1 µL of the GAM-CNP conjugate was added. This mixture was briefly vortexed and added to the sample well of the LFD. The LFIA was allowed to develop for 10 min. After visual readout of the result, the strips were removed from the LFD cassettes, sample- and wicking pad were removed, and then digitally analyzed in the Cube reader. Both the LFDs and the separate strips were photographed.

### 2.7. Application of the LFIAs to Surface Water Samples

To test the suitability of the newly developed indirect CHLP LFIA for application to environmental surface water samples, a CHLP stock of 200 µg/mL was spiked to a range of surface water samples (Appendix A) by adding 10 µL to 9990 µL water sample, resulting in a concentration of 200 µg/L. From here, a further dilution of the sample with sample water resulted in samples with 20 and 2 µg/L concentrations. Non-spiked water samples were used as negative controls. Samples were diluted 1:1 with 2 times concentrated RB (2× RB), composed of 200 mM BB pH 8.8, 2% BSA (*w*/*v*) and 0.01% Tween-20 (*v*/*v*). Portions of 98 µL of the samples were supplemented with diluted CHLP mAb (1 µL, 1:3200, in RB) and GAM-CNP conjugate (1 µL) and briefly mixed. The sample was then added to the sample well of the LFD and allowed to develop for 10 min. After incubation, the strips were analyzed and recorded as described.

### 2.8. Preparation of Assay Tubes for On-Site Application

To facilitate future on-site testing with the developed indirect CHLP LFIA, dedicated assay tubes were prepared and tested. Therefore, assay reagents were dried in tubes, largely based on a method previously described by Koets et al. [49]. In short, glass tubes were blocked with RB (300 µL) and incubated at 37 °C for 2 h. Next, 25 µL of GAM-CNP conjugate, 25 µL of CHLP mAb (both in 2× RB) and 50 µL of 2× RB were added at the bottom of the tube and gently mixed. This 100 µL reagent solution was dried under a nitrogen flow for 30 min at RT. CHLP calibration standards of 10 and 100 µg/L were prepared in water, and 100 µL of each standard were added to the dried reagents at the bottom of the glass tube. The glass tubes were gently mixed by hand for a few seconds, placed in a tube rack and the LFIA strip was inserted and incubated for 10 min and the readout was performed as described.

## 3. Results and Discussion

### 3.1. Choice of Assay Format

The indirect competitive assay format chosen for the development of the CHLP-specific LFIA (Figure 1), is based on earlier work on the detection of sulphamethazine in urine [50]. It was observed that the indirect approach, in which the specific antibody was diluted and mixed with the nanoparticle-coupled anti-species antibodies, resulted in a higher sensitivity compared to the direct approach, where the specific antibodies were coupled directly to the nanoparticles. In the indirect format, the precise dosing of the specific antibody molecules is necessary to balance between a good signal on the test line in case the sample contains no hapten and a sensitive displacement of the label, i.e., a decrease of the test line signal at low hapten concentrations. In a recent publication, Majdinasab et al. [51] applied a similar indirect approach and also concluded that the indirect assay is more sensitive than the direct assay [51,52]. In the indirect assay format, it is critical to use as low concentrations as possible for the labelled antibodies, i.e., enough to get a clear test line in the absence of hapten and not too much in order to obtain a decrease in signal (test line intensity) when the hapten is present at low concentrations.

### 3.2. Conjugate, Membrane, and Buffer Selection

In the initial experiments, several membranes were tested for the CHLP LFIA. From these membranes, CN140 was chosen as the most suitable, based on sensitivity and running time (data not shown). To choose between CHLP-OVA and CHLP-BSA, both conjugates were manually spotted on the membranes in several concentrations, combined with a control spot of the DAG pAb of a fixed concentration to ensure the functionality of the test. Over 500 hand-spotted strips were tested by applying several dilutions of the antibodies in four different running buffers. A selection of those 500 tests can be found in Appendix A. It was concluded that CHLP-BSA was the most suitable conjugate when spotted at 0.125 mg/mL, obtaining the best results in combination with a 1:3200 times (1.7 µg/mL) dilution of the specific antibody with RB as the running buffer.

### 3.3. Sensitivity Determination of the Sprayed LFIAs

Using the determined optimal conditions, CHLP-BSA test lines were sprayed on the NC membranes at a conjugate concentration of 0.125 mg/mL, while the control line was sprayed with DAG pAb at 0.05 mg/mL. This DAG pAb concentration was chosen to obtain similar intensities for both the test and control line. Roughly 30 strips were cut from each assembled card. These strips were placed in the LFD cassettes and calibration standards were applied to the sample well and the results were recorded after 10 min. To check for variability, this experiment was performed three times on the same day with 2 h intervals between the experiments (Figure 2). A visual readout by the naked eye determined the average sensitivity of the test to be between 5 and 1 µg/L. Next, the strips were removed from their cassettes and the intensities of the test lines were measured with the Cube reader. The average test line intensities for each data point (*n* = 6) were plotted against the CHLP concentrations as dose–response curves based on four parameter logistics (Figure 3).

Based on the dose–response curve, GraphPad Prism calculated a half maximum inhibitory concentration (IC_50_) of 1.97 µg/L. The limit of detection (LOD) was calculated by the average of all test line readings for the blank sample (*n* = 6), subtracted by three times the standard deviation for those blank sample measurements (3 × SD). This resulted in an LOD of 1.1 µg/L. Another set-up, where the strips were first inserted in a CHLP mAb solution and then into the GAM-CNP solution did not improve the assay sensitivity. Decreased sensitivities were observed when the CHLP mAb and GAM-CNP were added to the constructed LFD as reagent pads. Produced LFIAs were stable for at least 6 months (data not shown). When the limit of detection of the CHLP LFIA, 1.1 µg/L, is compared to the LOD of a classical immunoassay format like ELISA [37], the sensitivity of the LFIA is only 10-fold lower than that of the ELISA. The same holds for the comparison of the LOD of the LFIA to the LOD of the current in-house LC-MS/MS method (0.1 µg/L). The new CHLP LFIA is 2-fold less sensitive compared to an electrochemical immunosensing method on a microfluidic chip (0.5 µg/L) [53]. However, compared to the LFIA, both ELISA and LC-MS/MS are more laborious and expensive methods that require specific apparatus and are lab-based. The chip method is also a more expensive method and only two times more sensitive (0.5 µg/L) compared to the CHLP LFIA (1.1 µg/L). However, a better comparison is to focus on detection methods that are also simple, easy to perform, cheap, and suited for on-site application. In that perspective, the newly developed LFIA is rather sensitive compared to an enzymatic paper-based device based on the lipase enzyme combined with smartphone readout, which showed an LOD of 65 µg/L [35]. When compared to an existing LFIA for the detection of CHLP, based on a direct format with gold reporter molecules, the newly developed indirect CHLP LFIA was slightly more sensitive (factor 10) [44].

### 3.4. Application of the CHLP to Surface Water Samples

A total of eight water samples collected from various water bodies were fortified with CHLP at 200, 20, and 2 µg/L. Next, these water samples were mixed with an equal volume of 2× RB and then applied to the LFD. After 10 min, the LFIA results were evaluated visually to determine the sensitivities. After that, the strips were removed from the LFD and the sample pads and wicking pads were removed from the strip before digital imaging (Figure 4). Visually, the sensitivities were set at 2 µg/L for each water sample (Appendix A), which was equal to the sensitivities obtained upon digital imaging of the LFIAs (Figure 5, Appendix A). In some cases, the naked-eye determination of 2 µg/L was challenging, as was the case for the water sample from the Seine river. However, the digital imaging for this particular sample showed a clear distinction between the Test to Control ratio (T/C) for 1 µg/L (1.764) and the T/C for 0 µg/L (2.051).

Decision levels on the presence of low CHLP contaminations in surface waters should be based on the T/C ratio of the digital readings preferably without the use of CHLP calibration standards before each measurement. Therefore, the lowest reading of the blanks (minus the SD, 1.821) are compared to the highest T/C values obtained with the highest fortified samples at 2 µg/L (plus the SD, 1.764). Based on these values, only the Seine River (PS) sample may be considered an outlier come in the screening of those samples. Further assay optimization may still improve the sensitivity, e.g., different CHLP antibodies and/or conjugates. However, for 10 µg/L, which is the set detection limit by the EU, the highest T/C reading (0.656) is highly distinctive from the lowest blank reading and therefore it can be concluded that the sensitivity of the present CHLP LFIA is already fit for purpose.

### 3.5. Application to Other Bee-Related Matrices

Water is an important resource for bees, next to pollen and nectar. Therefore, the CHLP LFIA was also applied to pollen and apple blossoms. To this end, pollen and apple blossoms were fortified with CHLP and allowed to dry for 2 h. After extraction with water, and application to the CHLP LFD, sensitivities of 10 µg/g were observed for these extracts by visual readout. Digital imaging did not show improved detection limits for apple blossom (Appendix A). Unfortunately, these obtained sensitivities do not comply with the detection limit set by the EU. Extractions with water/methanol solvents, instead of water, did not show any improvements. This means that future work for the application of the CHLP LFIA on these bee-related matrices should focus on a suitable extraction method for more efficient extraction of those matrices and that are not detrimental to the performance of the LFIA (e.g., introducing matrix effects). Nevertheless, high concentrations of CHLP, as a consequence of a recent illegal use, will still be detected by the current LFIA. Moreover, the additives present in commercial pesticide formulae for field applications may contribute to easier extraction of CHLP from flowers or leaves. Therefore, further validation should also include real formulations as manufactured by CHLP producers. Additionally, the CHLP LFIA was applied to a fresh lime tree honey, free of CHLP (<0.1 µg/L). After a 10-fold dilution of honey with tap water, it was fortified with CHLP at several concentrations. The results in Figure 6 show that, even though the test lines are more vague when compared to the test lines obtained by application on surface waters, a sensitivity of 2 µg/L can be achieved by visual readout, which relates to a 20 ng/g (20 ppb) contamination of honey.

### 3.6. Towards Improved Point-of-Need Detection

Today, there is a strong need for rapid testing of pesticides at the point of need. Adequate screening and measuring at the point of need will save valuable time in decision-making and lab workloads (as negative samples do not need to be sent to the laboratory for confirmation). Pesticide detection assays for point of need application come in many formats, e.g., enzymatic lab-in-a-syringe detection [54], screen-printed electrodes (SPEs) for electrochemical (bio)sensing [55], surface-enhanced Raman spectroscopy (SERS) [56], and portable optical biosensors [57], each having its own advantages and disadvantages. However, it does not get much simpler in both equipment and handling than an LFD. The present CHLP LFD could in principle be performed at the point of need; however, it still needs three pipetting steps using laboratory pipettes to add the sample buffer, CHLP mAb, and the GAM-CNP, respectively. Therefore we tried to introduce simplifications based on the work of Koets et al. [49]. To this end, we added all the reagents and the 2× RB buffer into a 5 mL tube that was pre-blocked with RB and allowed the water fraction to evaporate under a nitrogen flow. In an initial experiment, we manually spotted strips and dried them as described previously. To the dried reagent mix at the bottom of the tube, tap water fortified with CHLP (100 and 10 µg/L) was added and mixed by manually shaking the tube. The LFIAs were then inserted and allowed to develop. Fortified tap water samples showed less intense spots compared to the blank tap water sample, indicating that this simplified approach has potential for on-site qualitative CHLP detection by naked eye (Appendix A). The introduction of simple, but accurate, disposable micropipettes would be the final step in avoiding any laboratory pipet usage [58]. Additionally, the developed LFIA strip can easily be measured at the point of need with the handheld Cube reader, as it does not need to be removed from the LFD cassette when using the tube-based method. Besides the Cube, the future digital imaging of the LFIA strips can also be carried out by using smartphones combined with freely downloadable applications [59]. Moreover, the current single analyte LFIA could be further developed as a multiplex LFIA by adding other targets which are also harmful to the (bee) environment, e.g., fipronil and pyrethroids. Additionally, it is worthwhile to mention that honeybees (and their hive products) can serve as excellent bioindicators because of their ability to provide high-resolution information on the presence of agrochemicals in the environment, as well as the repertoire of simple assays that can be performed to assess the effects of agrochemicals on their physiology and behavior [60].

## 4. Conclusions

An indirect competitive LFIA, based on a GAM-CNP detector conjugate, was developed for the detection of CHLP. The LFIA proved to be effective for detecting CHLP at low concentrations in standards and in European surface water samples. No background interference was observed (even though the water samples were from very diverse sources). A visual reading of the LFIA at or below the EU imposed limit of 10 µg/L proved to be successful, which is economically beneficial since it resolves the need for a digital imaging device. Nevertheless, the readout by rapid digital imaging was able to slightly enhance the sensitivities in some cases, which prevents the need for a trained or scientific eye. Additionally, the digital imager may be programmed to decide which samples test positive or negative. Since the digital imager is handheld, it is easy to use at the point of need. Improving the simplicity of the CHLP LFIA by using sample tubes with dried reagents makes the LFIA more user-friendly and even more suited for the on-site screening of surface waters. Besides surface water and honey, other bee-related matrices also need to be considered. However, for these matrices either the CHLP extraction protocols need to be improved or the sensitivity of the CHLP LFIA needs to be improved, as the present study shows that for the detection of CHLP in pollen and apple blossom the sensitivity is not sufficient.

## Figures and Tables

**Figure 1 biosensors-12-00735-f001:**
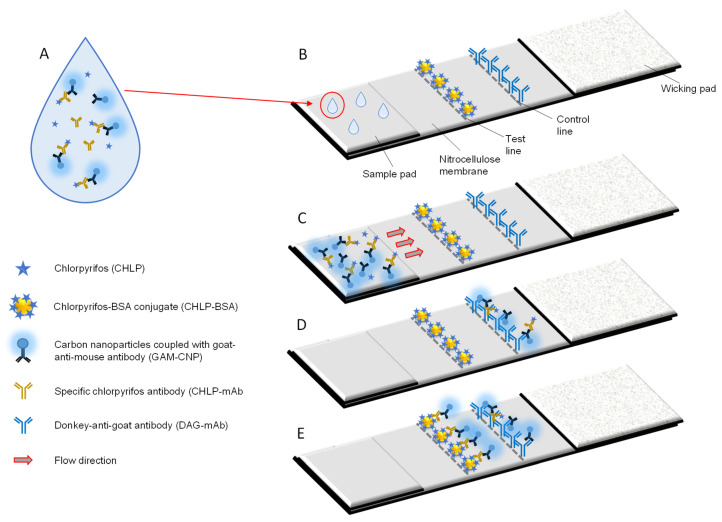
Principle of the CHLP lateral flow immunoassay (LFIA). Surface water samples are mixed with CHLP-mAbs and the GAM-CNP reporter (**A**). They are applied to the sample pad (**B**) and start to flow over the NC membrane towards the wicking pad of the LFIA (**C**). In case a high concentration of CHLP is present, the CHLP-mAbs will not bind to the CHLP-BSA conjugate on the test line. The GAM-CNP and the GAM-CNP/CHLP-mAb complex will bind to the DAG-mAb on the control line (**D**). In case no CHLP is present in the sample, the CHLP-mAb and the GAM-CNP/CHLP-mAb complex will bind to the CHLP-BSA conjugate on the test line and also to the DAG-mAb on the control line (**E**).

**Figure 2 biosensors-12-00735-f002:**
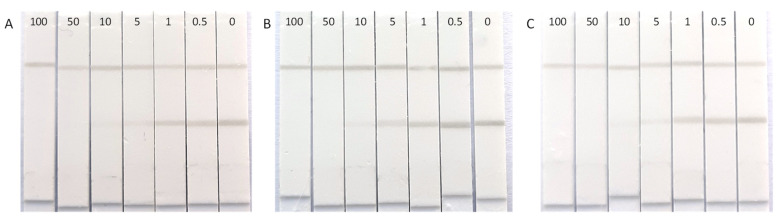
Sensitivity testing of the CHLP LFIAs by applying CHLP standards (0.5 to 100 µg/L) in three independent runs (**A**–**C**) within one day with a two hour interval between each run.

**Figure 3 biosensors-12-00735-f003:**
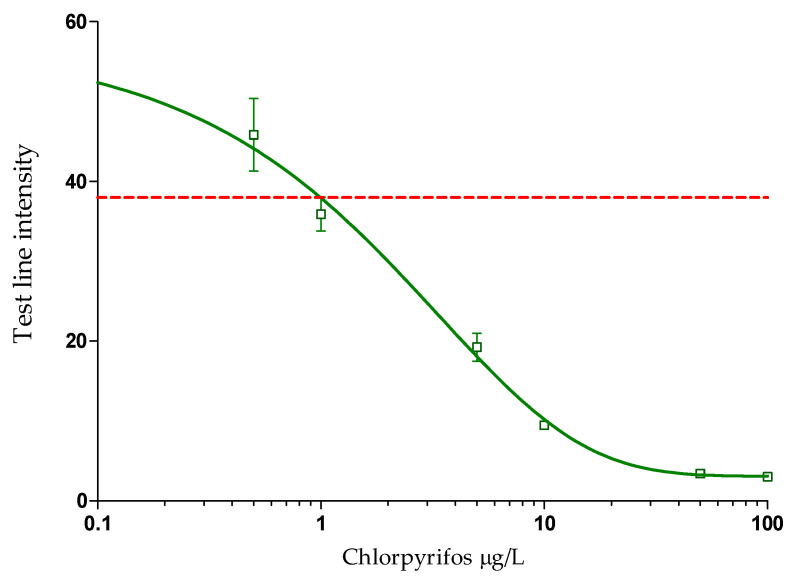
Dose–response curve constructed by four-parameter logistic regression, based on Cube readings (*n* = 2) of calibration standards (*n* = 3) in the CHLP LFIAs, performed in three independent experiments within one day with a two hour interval between each experiment, displayed as average test line intensities (*n* = 6). The calculated limit of detection is indicated by the red dotted line.

**Figure 4 biosensors-12-00735-f004:**
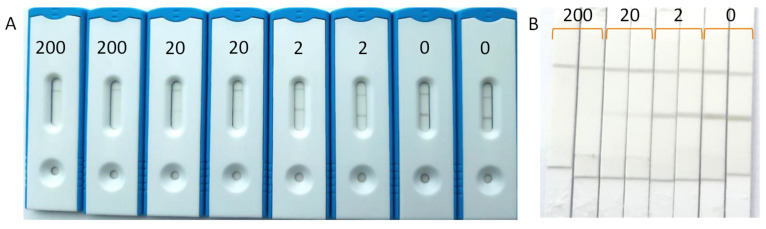
Surface water from a brook (Wageningen, The Netherlands) fortified with CHLP at 200, 20, 2, and 0 µg/L and applied to the LFIAs in duplicate. Visual imaging was performed in the assembled cassettes (**A**) and with strips removed from the cassettes for digital imaging (**B**).

**Figure 5 biosensors-12-00735-f005:**
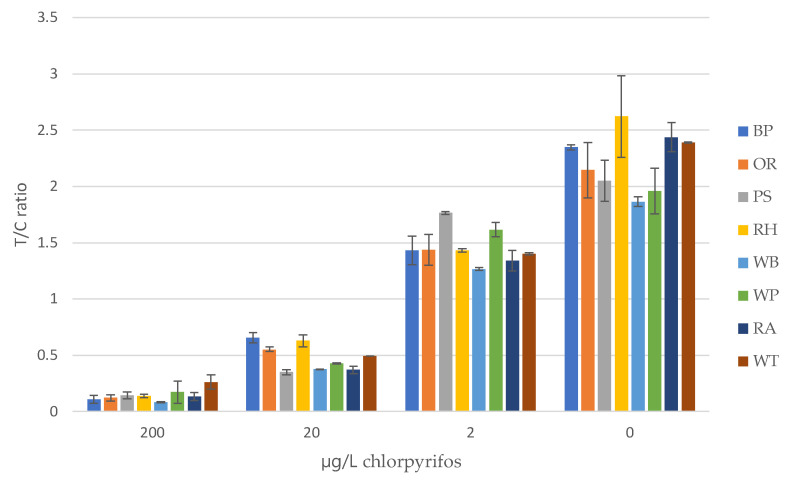
Average T/C ratios for all the CHLP fortified water samples. BP = Pond, Brussels; OR = Rhine, Oosterbeek; PS = Seine, Paris; RH = Harbor, Rotterdam; WB = Brooke, Wageningen; WP = Pond, Wageningen; AQ = aquaponics water, Rome; and T = tap water.

**Figure 6 biosensors-12-00735-f006:**
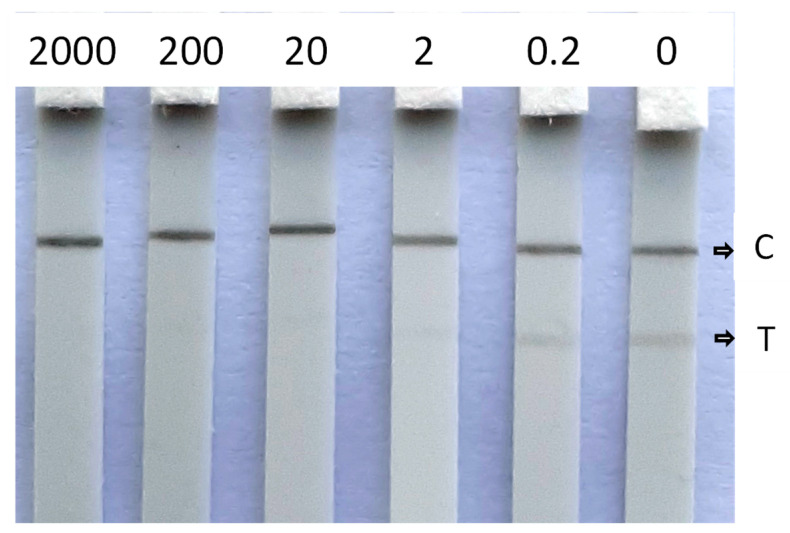
Visual detection of CHLP (µg/L) in fortified diluted fresh lime tree honey with C indicating the control line and T indicating the test line.

## Data Availability

The data presented in this study are available from the corresponding author upon request.

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
