# Peer review of "Biosensing Chlorpyrifos in Environmental Water Samples by a Newly Developed Carbon Nanoparticle-Based Indirect Lateral Flow Assay"

_biosensors, 2022, doi:10.3390/bios12090735_

Round 1

Reviewer 1 Report

The manuscript is adapted to the scope of the journal. The science is well thought out. The simplicity of the devices developed and the equipment involved could have a good impact if they were implemented at the point of use. As usual, there are a few things to keep in mind that authors need to address, the following main points:

 1.  I recommend adding information on point-of-use devices for pesticide determination in the introduction. A useful article for this is: doi:10.3390/bios10040032.    

2. Authors should add a table comparing the analytical characteristics of their device with those of other methods for pesticides  

3.  I suggest the comparison of the analytical characteristics (detection limit, linear range, precision) with a reference method for the detection of chlorpyrifos, such as HPLC-MSMS or ELISA. Add a detailed explanation of the experiment.  

4. Conduct a study of possible interference from other pesticides such as parathion, imidacloprid, glyphosate... adding references  

5. The authors should show the sigmoidal calibration graph of the lateral flow immunoassay and in an inset the interval lineal.     

6. Carry out the analysis of other types of more complex matrices such as fruit or honey, due to their importance in the death of bees. Add in a table the peak and recovery data of all the samples analyzed with the LFD.

Author Response

Response to Reviewer 1:

he manuscript is adapted to the scope of the journal. The science is well thought out. The simplicity of the devices developed and the equipment involved could have a good impact if they were implemented at the point of use. As usual, there are a few things to keep in mind that authors need to address, the following main points:

  1. I recommend adding information on point-of-use devices for pesticide determination in the introduction. A useful article for this is: doi:10.3390/bios10040032.

Response to comment 1: We agree with the reviewer and have therefore implemented the additions requested. Since the point of need detection was not the main scope of the manuscript, but rather an outlook, we decided to add this new section to chapter 3.6, including references to other recent methods developed and/or surveyed for their point of need applicability. 

  1. Authors should add a table comparing the analytical characteristics of their device with those of other methods for pesticides
  2. I suggest the comparison of the analytical characteristics (detection limit, linear range, precision) with a reference method for the detection of chlorpyrifos, such as HPLC-MSMS or ELISA. Add a detailed explanation of the experiment.  

Response to comments 2 and 3: 

We agree with the reviewer that a critical comparison is lacking and therefore have made a critical comparison to relevant current technologies, as well as, golden standards like ELISA and LC-MS/MS, and added it as text in chapters. We believe that compiling a detailed table is more relevant for a review paper on chlorpyrifos detection methods than for the current method development.

  1. Conduct a study of possible interference from other pesticides such as parathion, imidacloprid, glyphosate... adding references

We do agree with the reviewer that information on cross-reactivity is lacking in the current manuscript. Therefore we implemented cross-reactivity information of the antibody, as it was tested extensively in a research project where a multiplex microsphere immunoassay was developed for the detection of bee-harming pesticides. This work is currently written up for a peer-reviewed manuscript to be published in 2022. We added the following section to the current manuscript to introduce this information:

Cross-reactivity of other pesticides with the CHLP mAb were previously tested in a multi-plex microsphere immunoassay format within the H2020 B-GOOD  project, focusing on pesticides harmful to bees. In total 88 pesticides were tested from the pesticide classes neonicotinoids, pyrethroids, avermectins, carbamates, organophosphates and some single pesticides, including fipronil. From these pesticides, only dicrotophos, triazophos, para-thion and methyl-parathion showed significant cross-reactivity but only at relatively high concentrations (1000 µg/mL) (unpublished results).   

  1. The authors should show the sigmoidal calibration graph of the lateral flow immunoassay and in an inset the interval lineal.

The original graphs were calculated with 4 parameter logistics from separate experiments. To better show the variation in the measurements, we decided to replace figure 3 with a new one where all the separate data was integrated into one curve, expressing the test line intensities instead of the relative responses, including standard deviations (n=6). This also resulted in the definition of an IC50 value and a calculated limit of detection. However, within Graphpad Prism software there is no definition of a linear range as this is considered arbitrary by the developers (https://www.graphpad.com/guides/prism/latest/curve-fitting/reg_why-prism-doesnt-attempt-to-f.htm). From Figure 3, a linear range could be concluded by readers themselves if deemed necessary (between 10 and 1 ppb, by approach).

  1. Carry out the analysis of other types of more complex matrices such as fruit or honey, due to their importance in the death of bees. Add in a table the peak and recovery data of all the samples analyzed with the LFD.

We do agree with the reviewer that in the light of the European project in which this work was conducted, the importance of the CHLP LFD could be more dedicated to the health status index of bees. Initially, we applied the CHLP LFD to the main food sources of bees, water and pollen. At the reviewer’s request, we have applied the LFIA to spiked honey samples and added the successful results as a new figure (6) in the manuscript.

Reviewer 2 Report

Willemson et al. showed the detection of chlorpyrifos in water sample using the lateral flow assay based on carbon nanoparticles. Few points need to be considered:

1.     Author should provide the characterization of carbon nanomaterials used in the study.

2.     There is no evidence regarding the coating of antibodies over carbon nanomaterials.

3.     Limit of detection and linear detection range needs to be calculated.

4.     Selectivity of sensor is an important criteria, author should provide the selectivity data.

5.     Quantification of developed detection spot needs to be done.

6.     Comparative table for comparison of various developed sensors should be provided.

Author Response

Response to Reviewer 2

Willemson et al. showed the detection of chlorpyrifos in water sample using the lateral flow assay based on carbon nanoparticles. Few points need to be considered:

  1. Author should provide the characterization of carbon nanomaterials used in the study.

We agree with the reviewer that this info is lacking. The batch of carbon nanomaterials used in this study have been previously characterized, including a detailed SWOT analysis, and used ever since in numerous peer-reviewed publications. Those details can be found in the publication: Amorphous carbon nanoparticles: a versatile label for rapid diagnostic (immuno)assays and has now been added to the materials section.

  1. There is no evidence regarding the coating of antibodies over carbon nanomaterials.

We agree with the reviewer that the coating of the antibodies to the carbon nanomaterials is not controlled. However, it is a standard operating procedure within our institutes that is highly reliable and reproducible and was previously tested by flocculation strategies. The corresponding publication: Colloidal carbon particles as a new label for rapid immunochemical test methods: Quantitative computer image analysis of results, is now added as a reference to the method section of the manuscript.

  1. Limit of detection and linear detection range needs to be calculated.

We agree with the reviewer and have modified Figure 3 to make it better suitable for calculating a defined limit of detection and IC50. All the separate data was integrated into one curve, expressing the test line intensities instead of the relative responses, including standard deviations (n=6). This resulted in the definition of an IC50 value and a calculated limit of detection. This information was integrated into the manuscript. However, within Graphpad Prism software there is no definition of a linear range possible for sigmoid curves, as this is considered arbitrary by the developers (https://www.graphpad.com/guides/prism/latest/curve-fitting/reg_why-prism-doesnt-attempt-to-f.htm). From Figure 3, a linear range could be concluded by readers themselves if deemed necessary (between 10 and 1 ppb, by approach).

  1. Selectivity of sensor is an important criteria, author should provide the selectivity data.

We do agree with the reviewer that information on cross-reactivity is lacking in the current manuscript. Therefore we implemented cross-reactivity information of the antibody, as it was tested extensively in a research project where a multiplex microsphere immunoassay was developed for the detection of bee-harming pesticides. This work is currently written up for a peer-reviewed manuscript to be published in 2022. We added the following section to the current manuscript to introduce this information:

Cross-reactivity of other pesticides with the CHLP mAb were previously tested in a multi-plex microsphere immunoassay format within the H2020 B-GOOD  project, focusing on pesticides harmful to bees. In total 88 pesticides were tested from the pesticide classes neonicotinoids, pyrethroids, avermectins, carbamates, organophosphates and some single pesticides, including fipronil. From these pesticides, only dicrotophos, triazophos, para-thion and methyl-parathion showed significant cross-reactivity but only at relatively high concentrations (1000 µg/mL) (unpublished results).

  1. Quantification of developed detection spot needs to be done.

We agree with the reviewer that this could be of interest, however, the applied Cube reader is a line-based reader and cannot be applied as a spot reader. Nevertheless, visual detection shows that the spots are clearly varying in intensity, which shows the potential suitability for on-site testing with visual read-out.

  1. Comparative table for comparison of various developed sensors should be provided.

We agree with the reviewer that a critical comparison is lacking and therefore have made a critical comparison to relevant current technologies, as well as, golden standards like ELISA and LC-MS/MS, and added it as text in chapters. We believe that compiling a detailed table is more relevant for a review paper on chlorpyrifos detection.

Reviewer 3 Report

I recommend its publication with minor revision and re review as listed below.

1. Conjugation of GAM antibodies to amorphous CNPs- did varied the carbon ratio and did you did optimization, if yes please include data graphs

2. CHLP-OVA were diluted in PBS- what type of calculation involved means dilution factor

3. The sprayed NC was dried for 30 minutes—if drying time less and more, let me know the possibility

4. Sensitivity testing of the CHLP LFIAs- what is the error factor

5. The spiked water samples- explain with detail

6.  What about stability  and reproducibility

7. English needs more improvement in the manuscript.

8. The quality of given graphical abstract is low, author should improve.

9. Recheck all the abbreviations, equations, tables and figures (there captions).

10.  Analysis needs more clarification.

Author Response

Response to reviewer 3

  1. Conjugation of GAM antibodies to amorphous CNPs- did varied the carbon ratio and did you did optimization, if yes please include data graphs

We do agree with the reviewer that this is an interesting point. For this manuscript, the absorption of the GAM antibodies was only done at one concentration. For that, we followed a standard procedure that was previously successfully developed in the labs of co-authors van Amerongen en Wichers. This standard operating procedure within our institutes is highly reliable and reproducible and was previously tested by flocculation strategies. The corresponding publication: Colloidal carbon particles as a new label for rapid immunochemical test methods: Quantitative computer image analysis of results from 1993, is now added as a reference to the method section of the manuscript.

  1. CHLP-OVA were diluted in PBS- what type of calculation involved means dilution factor

We agree with the reviewer that dilution factors and calculations are not provided for this, as we think calculating and diluting is general laboratory practice. Nevertheless, we have adjusted this section  in the manuscript accordingly:

The stock solution of CHLP-BSA and the CHLP-OVA, both is PBS, were diluted with PBS (10 mM potassium phosphate buffer, pH 7.4) to obtain a concentration series of 1 mg/mL, 0.75 mg/mL, 0.5 mg/mL, 0.25 mg/mL and 0.125 mg/mL. To this end, the CHLP-BSA stock was diluted 5.7 times for a 1 mg/mL solution and 7.6 times for the 0.75 mg/mL solution. From the 1 mg/mL stock, 2-fold serial dilutions were made to obtain 0.5 mg/mL, 0.25 mg/mL and 0.125 mg/mL solutions . For CHLP-OVA, 2-fold serial dilutions were made directly from the 1 mg/mL stock and for the 0.75 mg/mL solution a 1.33-fold dilution was applied.

  1. Sensitivity testing of the CHLP LFIAs- what is the error factor

We agree with the reviewer and have modified Figure 3 to better show the error factor. To this end, all the separate data was integrated into one curve, expressing the test line intensities instead of the relative responses, including standard deviations (n=6). This also resulted in the definition of an IC50 value and a calculated limit of detection. This information was integrated into the manuscript.

  1. The spiked water samples- explain with detail

We have added additional information to this section to better explain the procedure.

  1. What about stability  and reproducibility

The stability of the LFIAs used for this manuscript is at least 6 months. We have added this detail to the manuscript.  Further storability testing is ongoing and can eventually shed a light on this question. In our labs, we have observed functionality of produced LFIAs, from 1 to 10 years. The produced carbon conjugate is generally stable for up to 6 - 24 months when stored at 4 degrees, this detail was also added to the manuscript. Concerning the reproducibility of the measurement, it is shown in figure 2 and 3 where we measured the same samples 3 times divided over the day. A new batch of LFIAs was tested for its sensitivity and applied for the successful detection of chlorpyrifos in honey, which was a request from another reviewer Those results were added to the manuscript.

  1. English needs more improvement in the manuscript.

As requested by the reviewer, we made English improvements throughout the whole manuscript.

  1. The quality of given graphical abstract is low, author should improve.

We are a bit puzzled by this remark, as we have not submitted a graphical abstract for this publication. If the reviewer means figure 1 of the manuscript, in our document it can be magnified up to 400% while still retaining its high quality (in the word file). The same holds for the separate TIF file that was submitted.

  1. Recheck all the abbreviations, equations, tables and figures (there captions).

As requested by the reviewer, we have rechecked all the abbreviations, equations, tables, and figure captions and made changes to benefit the consistency of the manuscript.

  1. Analysis needs more clarification

We agree with the reviewer and have modified the data analysis part by introducing a new figure 3 and accompanying text. Both the IC50 and the limit of detection were calculated. Additionally, changes were made to the manuscript for more clarification in general.

Round 2

Reviewer 1 Report

I consider that the article is suitable for publication in the journal Biosensors.

Author Response

We are grateful to reviewer 1 for the acceptance of the manuscript.

Sincerely,

Jeroen Peters

Reviewer 2 Report

Comment 1: Authors were asked to provide the characterization data, but only other paper reference has been provided in response file. As written by authors “Amorphous carbon nanoparticles: a versatile label for rapid diagnostic (immuno)assays has now been added to the materials section” I can not find it in the revised manuscript. Authors can provides exact location with line number.

Comment 2: I am not able to locate the “Quantitative computer image analysis of results” in revised manuscript as stated by authors. Authors can provides exact location with line number.

Comment 4: Again in comment 4, other project/paper has been cited no data provided.

Comment 5: Author stated that “visual detection shows that the spots are clearly varying in intensity” but from visual detection it is not possible to quantify the result.

Comment 6: Authors stated that “We believe that compiling a detailed table is more relevant for a review paper on chlorpyrifos detection.” However, comparative table was asked to provide the high efficiency and benefits of proposed sensor over the already reported sensor.

Round 3

Reviewer 2 Report

Accept